# Development and Feasibility of an Inpatient Cancer-Related Sarcopenia Pathway at a Major Cancer Centre

**DOI:** 10.3390/ijerph19074038

**Published:** 2022-03-29

**Authors:** Jenelle Loeliger, Lara Edbrooke, Robin M. Daly, Jane Stewart, Lucy Bucci, Carmen Puskas, Marnie Fitzgerald, Brenton J. Baguley, Nicole Kiss

**Affiliations:** 1Nutrition & Speech Pathology Department, Peter MacCallum Cancer Centre, Melbourne, VIC 3000, Australia; jane.stewart@petermac.org (J.S.); carmen.puskas@petermac.org (C.P.); 2Department of Health Services Research, Peter MacCallum Cancer Centre, Melbourne, VIC 3000, Australia; lara.edbrooke@petermac.org; 3Physiotherapy Department, The University of Melbourne, Parkville, VIC 3010, Australia; 4Institute for Physical Activity and Nutrition, Deakin University, Geelong, VIC 3220, Australia; robin.daly@deakin.edu.au (R.M.D.); b.baguley@deakin.edu.au (B.J.B.); nicole.kiss@deakin.edu.au (N.K.); 5Physiotherapy & Occupational Therapy Department, Peter MacCallum Cancer Centre, Melbourne, VIC 3000, Australia; lucy.bucci@petermac.org (L.B.); marnie.fitzgerald@petermac.org (M.F.)

**Keywords:** sarcopenia, low muscle mass, cancer, nutrition, exercise, care pathway, multimodal, malnutrition

## Abstract

Cancer-related sarcopenia is a complex condition; however, no cancer-specific clinical model is available to guide clinical practice. This study aims to (1) develop an evidence-based care pathway for the management of cancer-related sarcopenia (“sarc-pathway”) and (2) pilot test the feasibility (reach, intervention fidelity, patient and clinician acceptability) of the sarc-pathway in an inpatient cancer ward. The sarc-pathway was developed using a care pathway format and informed by the current literature. Patients admitted to a 32-bed inpatient cancer ward were recruited to receive sarc-pathway care and the feasibility outcomes were assessed. Of the 317 participants admitted, 159 were recruited over 3.5-months (median age 61 years; 56.0% males). Participant consent was high (99.4% of those approached) and 30.2% were at risk of/had sarcopenia. The sarc-pathway screening, assessment and treatment components were delivered as intended; however, low completion of clinical assessment measures were observed for muscle mass (bioimpedance spectroscopy, 20.5%) and muscle function (5-times chair stand test, 50.0%). The sarc-pathway was demonstrated to be acceptable to patients and multidisciplinary clinicians. In an inpatient cancer ward, the sarc-pathway is a feasible and acceptable clinical model and method to deliver and adhere to the sarcopenia clinical parameters specified, albeit with further exploration of appropriate clinical assessment measures.

## 1. Introduction

Sarcopenia is a serious condition associated with ageing that is characterised by a loss of skeletal muscle mass and strength, with many adverse consequences on the physical function and health-related quality of life [1,2]. The European Working Group on Sarcopenia in Older People 2 (EWGSOP2) recognises that sarcopenia can also be disease-related, which is known as secondary sarcopenia and commonly occurs in conditions such as organ failure or cancer [2]. Studies investigating cancer-related sarcopenia primarily refer to sarcopenia as muscle loss alone. Depending on cancer type and treatment, cancer-related sarcopenia can occur in up to 60% of patients [3,4,5]. Cancer-related sarcopenia is a multifaceted condition often occurring alongside cancer-related malnutrition, and has a complex underlying pathology [2].

Evidence-based practice guidelines and models of care surrounding the identification, assessment and management of cancer-related sarcopenia are less well advanced when compared to malnutrition [2,6,7,8]. Whilst there are similarities between these two conditions, there are distinct valid and reliable screening and assessment tools for each. To help guide suitable clinical care, a cancer-related malnutrition and sarcopenia position statement developed by the Nutrition Group within the Clinical Oncology Society of Australia (COSA) was released in 2020 [6]. This position statement aims to guide and support evidence-based clinical practice in relation to both cancer-related malnutrition and sarcopenia. This position statement is particularly useful for the clinical guidance of suitable tools for the screening and assessment of cancer-related sarcopenia and potential strategies for effective interventions, as evidence continues to emerge in this area [6].

The implementation of evidence-based malnutrition management has been a focus over the past 15 years at our cancer centre, including the clinical integration of valid and reliable screening, assessment tools and tumour stream care pathways that provide a clinical framework for effective and timely nutrition care [9,10,11,12]. Current practice at our cancer centre does not include any systematic and/or an evidence-based clinical approach for the identification, assessment and management of sarcopenia in patients with cancer and no relevant policies or clinical guidelines are utilised to guide its management. Therefore, this presents evidence of a practice gap at our cancer centre.

The use of Allied Health Assistants (AHAs) in hospitals to support allied health professionals and assist in providing therapeutic care to patients is well established [13]. Having been part of our cancer centres allied health workforce for over 10 years, the AHA’s work across multiple disciplines and play a key role in many tasks, including risk screening, basic assessments and treatments, data collection and equipment provision. The AHA workforce is a suitable workforce to take on the role of screening for sarcopenia and triaging for further assessment. Care pathways provide an evidence-based framework to designate the actions and treatment that patients should receive at specified time intervals [14]. Care pathways assist in the translation of evidence-based practice at a practical level and can lead to care standardisation, reduction in practice variation and improvements in patient care, safety and outcomes [14,15].

This study aims to (1) develop an evidence-based care pathway for the management of cancer-related sarcopenia (“sarc-pathway”) and (2) pilot test the feasibility (reach, intervention fidelity and patient and clinician acceptability) of the sarc-pathway in a single inpatient cancer ward. Dependent on this study’s findings, the next step is to modify necessary components of the sarc-pathway and test the intervention effectiveness in a large-scale randomised controlled trial.

## 2. Materials and Methods

### 2.1. Study Design

This was a prospective mixed-methods single-group study. The study was conducted over two phases; (I) development of the cancer-related sarcopenia pathway (“sarc-pathway”) and (II) pilot testing the feasibility of the sarc-pathway. This study has been reported according to the Strengthening the Reporting of Observational Studies in Epidemiology (STROBE) statement to assist in quality reporting [16]. Ethical approval was received from the Peter MacCallum Cancer Centre Human Research Ethics Committee (HREC/75170/PMCC-2021).

### 2.2. Setting

This study was conducted within a single 32-bed inpatient ward within a single tertiary specialist cancer hospital in Melbourne, Australia. This inpatient ward consists of predominantly medical oncology patients who are a heterogeneous cohort in regard to cancer diagnoses and reasons for admission.

### 2.3. Participants

All patients admitted to the ward during a 3.5-month period (August–November 2021) were considered eligible for inclusion in the study. Non-English-speaking patients, those with impairment who could not give verbal consent and those receiving end-of-life care, were excluded. Eligible patients were approached and asked to consent verbally to receive clinical care, as described in the sarc-pathway.

### 2.4. Phase I: Development of the Sarc-Pathway

The project team aimed to design a tailored, evidence-based multidisciplinary sarcopenia management pathway including components for the identification, assessment and treatment appropriate for clinical care, which was based on published literature and following the recommendations from the COSA Cancer-related malnutrition and sarcopenia position statement [2,6,7,17,18]. The project team consisted of four nutrition (accredited practising dietitians) and four exercise oncology and sarcopenia experts (physiotherapists and an exercise physiologist) within clinical and academic settings, one clinical allied health assistant–nutrition assistant (AHA-NA) and included four authors of the COSA cancer-related malnutrition and sarcopenia position statement. During the planning of the study, consultation with a health economist and an expert in implementation science occurred to assist in the study design and frameworks utilised. The project team met approximately fortnightly to oversee the project and make decisions about the clinical and operational design components of the sarc-pathway. In addition, broader input on the design of the pathway was also sought from other cancer-specific allied health, such as nursing and medical staff working in the inpatient ward where the pilot was conducted.

#### 2.4.1. Adherence to Best Practice Audit

A pre-pilot adherence audit of current clinical practice (January–May 2021) in regard to cancer-related sarcopenia identification, assessment and treatment on the inpatient ward was conducted against key recommendations and practice tips from the COSA Cancer-related malnutrition and sarcopenia position statement [6]. The action, actor, context, target and time (AACTT) framework was utilised to clearly define the relevant criteria against each recommendation and practice tip to the inpatient ward context [19]. Three senior clinical operational leaders and members of the project team (J.L., J.S. and L.B.) agreed by consensus on whether clinical practice on the pilot ward either met (occurs ≥ 80% of the time), partially met (occurs ≥ 50–79% of the time) or did not meet (occurs < 50% of the time) each recommendation and practice tip from the position statement. This audit was repeated at the end of the pilot in Phase II based on clinical practice within the final one month of the pilot (November 2021).

#### 2.4.2. Developing Key Components of the Sarc-Pathway Model


For the overall sarc-pathway model, small changes were made as part of an iterative and continuous improvement cycle early in the pilot period and to enable compliance with local COVID-19 restrictions, i.e., with no group exercise classes available. Fortnightly project team meetings provided an avenue for decisions or changes to be made relating to the sarc-pathway over time. Appendix A describes the sarc-pathway model applied into clinical practice from approximately 3 weeks into the pilot until pilot end.The adherence to best practice audit helped identify local strengths and opportunities for improvement of the current clinical pathway.A clinical care pathway format was utilised to enable clear designation of actions at specific time-points [14,15] and to enable a comprehensive design; this was overlaid with the AACTT framework to help define behaviours and components of care to help assess uptake and/or adherence [19]. The component behaviours of action (behaviour), actor (who does the action), context (the setting in which the behaviour occurs), target (whom the actor performs the action on) and time (the time period or duration) were defined [19] (Table 1).Project team and wider multidisciplinary clinician input was sought and incorporated into the development of the sarc-pathway prior to Phase II piloting. The inclusion of valid and reliable screening tools, assessment and outcome measures into the sarc-pathway was considered essential. A pragmatic approach in the choice of tools and measures was taken, specifically in regard to equipment that was available to clinicians, the complexity and therefore the training required to conduct the measure and the appropriateness for the acute ward context. A two-page fact sheet specific to cancer-related sarcopenia and the sarc-pathway model was developed in consultation with patients and incorporated as part of the sarc-pathway screening process.The definition of sarcopenia was critical to defining the clinical criteria required for a sarcopenia diagnosis and the EWGSOP2 definition was chosen for the sarc-pathway [2]. The EWGSOP2 defines probable sarcopenia as low muscle strength and additionally, a sarcopenia diagnosis is confirmed if low muscle quantity (mass) is found [2]. All cut-points applied for each clinical measure are specified in Appendix A.Strong consideration was given to incorporation of the sarc-pathway into existing ward clinical processes, specifically in regard to risk screening and referrals to multidisciplinary allied health clinicians. AHA-NA were considered well placed to conduct screening in a ward setting, whilst the dietitians and physiotherapists were identified as the core expert clinical disciplines to provide nutrition and exercise assessments and interventions, respectively (as a multimodal package), to address cancer-related sarcopenia.


#### 2.4.3. Key Implementation Strategies Prior to Pathway Piloting


*Engaged project team*: The multidisciplinary and expert project team members met fortnightly throughout Phases I-II and applied a continuous quality feedback cycle to make real-time modifications to the sarc-pathway before and during the pilot.*Staff training and competency packages*: Tailored education and training was developed by members of the project team and provided to all multidisciplinary allied health clinicians providing care on the sarc-pathway. Competency packages for new tools or skills included in the sarc-pathway, not already used within usual clinical practices, were developed and completed by clinicians, specifically in regard to sarcopenia screening for the AHA-NA, body composition measurement via bioimpedance spectroscopy (BIS) for the dietitians and AHA-NA’s and five-time chair stand test (5-CST) for the physiotherapists and allied health assistant-physiotherapist (AHA-PT’s). Staff training and competency packages took 1–2 h to complete and were completed by staff within one month of commencing the pilot.*Embedding the sarc-pathway within digital technology*: Outcome measures were captured into discrete and reportable data fields and template documentation for all allied health clinician encounters for consistent reporting within the recruitment sites’ existing electronic medical record (EMR).*Communication*: Sharing of important milestones and key project updates occurred within allied health and with multidisciplinary nursing and medical colleagues approximately monthly throughout the project.


### 2.5. Phase II: Pilot Test the Feasibility of the Sarc-Pathway

The pilot of the sarc-pathway occurred over a 3.5-month period (August–November 2021). The pilot duration was determined by a number of factors, including availability of staffing resources, projected numbers of eligible patients and the current COVID-19 pandemic response within the hospital. All clinical and operational data were entered approximately weekly from the electronic medical record into the Research Electronic Data Capture (REDCap) secure web platform [20] by a project team member (CP).

#### 2.5.1. Reach

The number of patients (a) admitted during the pilot, (b) who met eligibility criteria, were approached and provided consent to receive care on the sarc-pathway and (c) received care on the sarc-pathway; were collected by the AHA-NA as part of usual clinical care processes, along with participant demographic, clinical characteristics and hospital length of stay.

#### 2.5.2. Intervention Fidelity (Intervention Delivery and Adherence)

Fortnightly project team meetings throughout the pilot enabled pragmatic and iterative changes to the sarc-pathway and/or clinical practices. Eighty to 100% represented high fidelity, 50–79% moderate fidelity and <50% low fidelity [21]. The following data were collected to measure intervention delivery and adherence:Number and proportion of participants screened as being ‘at risk/probable’ for sarcopenia who were referred to the dietitian and physiotherapist for assessment, based on the sarcopenia screening tool and calf circumference [combined strength, assistance in walking, rise from chair, climb stairs and falls (SARC-F) and calf circumference, SARC-CalF] [22] and handgrip strength (HGS) [23]Number and proportion of participants screened as being ‘at risk/probable’ for sarcopenia who were referred to the dietitian and physiotherapist and were assessed and received treatment, as indicated in the sarc-pathwayNumber and proportion of participants who had clinical assessment measures performed as per the sarc-pathway for nutritional status, muscle mass, muscle strength and/or physical function. These included:
○Nutritional status using the Patient-Generated Subjective Global Assessment (PG-SGA) [24,25] (performed by dietitian): this includes a subjective assessment of weight-loss, nutritional symptoms, food intake and activity levels and an objective assessment of body composition (fat, muscle stores and fluid status, scored as “0” = no deficit, “1” = mild deficit, “2” = moderate and “3” = severe). Each component of the PG-SGA is scored between 0 and 4 to provide an overall score (typical scores range from 0 to 35) and category of nutritional status (A = well-nourished, B = moderately/suspected malnutrition and C = severe malnutrition);○Muscle mass using (i) BIS for estimated appendicular lean mass (ALM) [2] (performed by dietitian): segmental analysis on the Impedimed SOZO^TM^ estimates ALM equating skeletal muscle mass of each arm and leg [26,27]; (ii) Body Mass Index (BMI)-adjusted calf circumference as a proxy measure of muscle mass (performed by AHA-NA) [2,28].○Muscle strength using (i) the 5-times chair stand test (5-CST) [2,29] (performed by physiotherapist) measures the time a participant takes to stand up and sit down 5 times, without using arms, from a standard height chair; and (ii) hand grip strength (performed by AHA-NA), utilising the Jamar dynamometer with clinicians recording 3 measurements on each side from the participant and using the maximum result [2,23].○Performance status using the Australia-modified Karnofsky Performance Status scale (AKPS) [30] (performed by physiotherapist): utilised to measure overall performance status, whereby the clinician observes the participant’s ability to perform common tasks relating to activity, work and self-care. It is assessed on an 11-point scale with a higher score equating to a better level of function, ranging from 0 (dead) to 100 (normal and no complaints; no evidence of disease).

#### 2.5.3. Participant and Multidisciplinary Clinician Acceptability

Acceptability of the sarc-pathway was measured using purpose-designed surveys based on key dimensions of acceptability, as described by the theoretic framework of acceptability (TFA) (Appendix A) [31]. The TFA comprises seven domains designed to assess multifaceted healthcare intervention acceptability elements (affective attitude, burden, ethicality, intervention coherence, opportunity costs, perceived effectiveness and self-efficacy) [31]. Participants eligible to complete the acceptability survey had completed the screening component of the sarc-pathway at a minimum. All multidisciplinary allied health clinicians who had applied the sarc-pathway as part of their clinical care practices during the pilot period were invited to participate in the clinician acceptability survey. Both surveys asked participants to rate against each TFA domain on a 5-point Likert scale from ‘strongly disagree’ to ‘strongly agree’ and included the options for free-text comments. Ten surveys were selected as a target sample size for participants, based on pragmatic restrictions on the project team members’ abilities to distribute surveys who were not clinicians involved in their care and were distributed during the final two weeks of the pilot. Multidisciplinary clinician surveys were distributed at the end of the pilot to any AHA-NA, dietitian, physiotherapists or AHA-PT who underwent staff training and provided clinical care on the sarc-pathway for four or more weeks during the pilot period.

#### 2.5.4. Exploratory Outcomes

The prevalence of patients screened as ‘at risk/probable’ for sarcopenia and those subsequently diagnosed with sarcopenia was determined. In addition, the prevalence of patients screened as ‘at risk’ of malnutrition (Malnutrition Screening Tool (MST), score of ≥2) [32] was determined from usual clinical practices.

#### 2.5.5. Resource Utilisation Associated with Implementation of the Sarc-Pathway

The time taken for (a) the AHA-NA to conduct screening and make relevant referrals and for (b) the dietitians and physiotherapists to conduct assessments and provide treatment to participants was recorded in the EMR in real-time.

### 2.6. Data Management and Analysis

Clinical and operational data were entered from the EMR into the database (REDCap). To summarise the variables, descriptive statistics were reported as means and standard deviations or medians and interquartile ranges for continuous variables, as appropriate dependent on normality of data distributions. Categorical variables and adherence to best practice audit results were reported as counts and proportions.

Completion of the clinician and participant acceptability surveys was voluntary and consent was implied through completion of the anonymous survey by participants. Participant acceptability surveys were completed in hard copy. Clinician acceptability surveys were completed directly into REDCap via an email link sent to all eligible clinicians. All quantitative data were analysed descriptively and reported as frequencies, percentages, means or medians. Open-ended questions were analysed qualitatively using content analysis to identify key themes in the responses.

## 3. Results

### 3.1. Characteristics of Participants and Reach

Three-hundred and seventeen participants were admitted to the ward for ≥1 night during the 3.5-month pilot period and 160 (50.5%) participants were eligible and able to be approached for sarcopenia screening (Figure 1). One-hundred and fifty-seven patients (49.5%) were either ineligible or not approached by the AHA-NA for sarcopenia screening; the reasons for this were not recorded. One-hundred and fifty-nine (50.2%) of those admitted consented to care on the sarc-pathway, representing 99.4% of those eligible and approached.

The median age of participants was 61 years, 56.0% were male and the most common diagnosis of this cohort was sarcoma (17.0%) followed by lung cancer (16.4%) (Table 2).

### 3.2. Intervention Fidelity (Intervention Delivery and Adherence)

#### 3.2.1. Intervention Delivery

The COVID-19 pandemic response did impact on the commencement date of the pilot and led to mitigation strategies, such as modification within the sarc-pathway design (such as no exercise classes), to enable the study to proceed [33]. Table 3 and Figure 2 show the adherence to the planned delivery of the key components of the sarc-pathway, specifically in relation to ‘time’ (as described in Table 1). Moderate adherence (74.2%) was achieved in meeting the screening timeframes specified in the sarc-pathway i.e., within 2 days of admission. High adherence was achieved by the multidisciplinary team in delivering the key components relating to referrals (100% within timeframes) and assessment and treatment (100% and 91.7% within timeframes, for the dietitian and physiotherapist, respectively). All ‘actions’ described within the sarc-pathway (as per the AACTT framework) were delivered as intended except for clinical assessment methods (described below), and all other component behaviours of the ‘actor’ (who does the action), ‘context’ (the setting in which the behaviour occurs) and ‘target’ (whom the actor performs the action on) were delivered as intended.

#### 3.2.2. Clinical Assessment Measures

The screening clinical assessment measures were conducted by the AHA-NA on all participants that consented to screening (*n* = 159), indicating high fidelity in the delivery and adherence to the initial screening process overall (Table 4). Five of 17 (29.4%) participants who had a ‘not at risk’ screening result on admission were re-screened after 7 days (+/− 1 day) if they remained an inpatient, and 12 did not get re-screened, as they were missed by the AHA-NA.

The clinical assessment measures conducted for those participants considered ‘at risk/probable’ for sarcopenia by the dietitian and physiotherapist had variable completion rates. The dietitian was able to measure estimated ALM (via BIS) for 20.6% of participants and 76.5% participants had a PG-SGA completed, whilst the physiotherapist completed the 5-CST on 50% and the AKPS on 72.2% of the assessed participants (Table 4). The reasons for non-completion of the ALM measure were primarily related to time and equipment burden for both the participant and clinician. This burden included difficulty in access and transport of the equipment (located off ward; therefore, transport of the equipment to/from ward was required and the setup was time consuming), transfer of the patient to/from the equipment and using the device was time consuming and difficult for fatigued patients or those with mobility issues.

#### 3.2.3. Pre-and Post-Adherence to Best Practice Audit

An improvement was demonstrated following the implementation of the sarc-pathway in the ability to meet recommendations and practice tips stated in the cancer-related malnutrition and sarcopenia position statement, with only 2 of 15 pre-pilot recommendations and practice tips met or partially met, as compared to the 15 of 15 post-pilot (Appendix A) [6]. For every component of the audit, consensus was unanimous against each recommendation and practice tips, with no disagreements between the raters. Furthermore, all recommendations and practice tips positively changed between the pre-pilot and post-pilot (except for one recommendation that was already met pre-pilot) (Appendix A).

### 3.3. Participant and Multidisciplinary Clinician Acceptability

#### 3.3.1. Participant Acceptability Survey

Ten participants were approached and seven surveys were completed (70% response rate) within the final 3 weeks of the pilot. Overall, 100% of participants who completed the survey and were seen by the AHA-NA (*n* = 7), dietitian or physiotherapist (*n* = 4) either agreed or strongly agreed that they were ‘overall satisfied with their session’ with the respective allied health professional. Participant acceptability was deemed high for all domains of the TFA, as indicated by 100% of the responses being slightly agree, agree or strongly agree for those framed in the positive (or strongly disagree and disagree for those framed in the negative) (Figure 3a,b) [31].

#### 3.3.2. Clinician Acceptability Survey 

Twelve surveys were distributed to clinicians and 12 completed (100% response rate) in the final week of the pilot. Two AHA-NA’s, four dietitians, five physiotherapists and one AHA-PT completed the survey, with a varied number of years of experience in their respective professions (33.3% <1 year, 33.3% 1–5 years, 33.3% 6–10 years). The majority (66.7%) of the 12 multidisciplinary clinicians either strongly agreed or agreed that ‘the introduction of the sarc-pathway into clinical care improves the screening, assessment, diagnosis and intervention of sarcopenia for patients’ (Figure 4a,b). Clinician acceptability was deemed high for the TFA domain asking ‘fits with my personal values’ (100% of responses being slightly agree, agree or strongly agree), however there were variable responses in regard to acceptability for the remaining six TFA domains. The AHA-NA’s and physiotherapists (including the AHA-PT) reported moderate acceptability across most TFA domains, however the dietitians reported low acceptability primarily in regard to time/effort, enjoyment and seeing benefits for patient outcomes and care.

### 3.4. Exploratory Outcomes

On admission screening, 30.2% (*n* = 48) of the study sample were considered ‘at risk/probable’ for sarcopenia (Table 5). Following assessment by the dietitian, only 8.8% (*n* = 14) were diagnosed with sarcopenia according to the clinical criteria applied in the sarc-pathway, of which half (*n* = 7) were using criteria inclusive of ALM with the remainder applying the specified proxy measure, the BMI-adjusted calf circumference, as an ALM measure was not available. Malnutrition risk was observed in 38.8% of the entire cohort and 26 participants were both ‘at risk’ of sarcopenia and malnutrition (54.2% of the ‘at risk/probable’ sarcopenia cohort). The median length of hospital stay was 5 days, indicating a relatively rapid turnover of participants on the pilot ward.

### 3.5. Resource Utilisation with Implementation of the Sarc-Pathway

The median time for the AHA-NA to complete the entire screening process per participant was 15 min (IQR 15, 20). The median time for the dietitian and physiotherapist to conduct a single assessment and treatment per participant was 45 min (IQR 30, 50) and 30 min (IQR 25, 40), respectively. A wide variation was found in the number of assessments/treatments required per participant under both the dietitian [median 0.0 (IQR 0, 2), range 0–21] and physiotherapist [median 0.0 (0, 2), range 0–31], noting that the majority of participants had a small number of assessments/treatments and those with a longer hospital length of stay were in the minority, with a higher number of assessments/treatments. Therefore, for an inpatient admission with a median hospital length of stay of 5 days, the estimated dietitian and physiotherapist resources to deliver assessments/treatments on the sarc-pathway were 61 min and 51 min per participant, respectively. During the pilot period, the dietitians and physiotherapists provided clinical care to a total of 72 (45.3%) and 69 (43.4%) participants, respectively of the study cohort (*n* = 159), as a result of either being on the sarc-pathway or for other assessment and treatment indications.

## 4. Discussion

Cancer-related sarcopenia is a complex condition and requires a multidisciplinary approach to its management to enable provision of high-quality care to patients. The sarc-pathway model of intervention developed and piloted in this study is, to our knowledge, the first of its kind to propose a clinically tailored pathway for an acute cancer inpatient ward that aims to support the adoption of guideline and position statement recommendations into clinical practice in addressing the issue of cancer-related sarcopenia. The sarc-pathway was comprehensively developed from published evidence and recommendations, applying the utility of a care pathway and component behaviours in accordance with the AACTT framework. This study has demonstrated the sarc-pathway to be a feasible method in which to reach the target audience and deliver and adhere to the sarcopenia clinical parameters specified, albeit with opportunities for exploration and improvement in regard to the choice of clinical assessment methods. The sarc-pathway achieved high acceptability to participants and moderate acceptability to multidisciplinary clinicians. Staff resources need to be considered for the implementation of a model, such as the sarc-pathway, into clinical practice.

Research into cancer-related sarcopenia in a real-world acute setting presented many challenges. The proportion of patients admitted to the ward that were approached to participate in this study (50.5%) was higher than in a recent feasibility study on sarcopenia in an acute ward with older adults (26.8%) [34]. Due to reduced resources during COVID-19 restrictions, the reasons for why 49.5% (*n* = 157) of participants were not approached in this study were not recorded. It is reasonable to propose that a small proportion of the admitted participants to the ward were ineligible (as few exclusion criteria were applied) and the most likely reason for patients not being approached by the AHA-NA was limited staff resourcing during the COVID-19 restrictions. As this was an unfunded study, limited AHA-NA staffing levels on the ward (and therefore competing priorities and tasks for the AHA-NA) impacted on the feasibility of implementing the sarc-pathway in a real-world setting. The screening process by the AHA-NA included gaining initial participant consent, completing each of the screening measures and explaining the results, providing simple education on sarcopenia and gaining consent for and referring participants ‘at risk of/probable’ for sarcopenia to the dietitian and physiotherapist. The AHA-NA screening tasks were modified early in the pilot period (as decided by the project team) due to the time required, limiting their ability to screen all eligible patients. The modification involved the BIS and 5-CST measures being included in the dietitian and physiotherapist assessments respectively, rather than with the AHA-NA, as initially piloted. Given the average time taken to complete the screening and referral process (15 min), consideration should be given to the benefit versus burden in identifying patients at risk of sarcopenia and whether or not a shorter or modified screening process could be an effective alternative.

The introduction of all screening measures (hand-grip strength, SARC-F and calf circumference) was successfully integrated into practice, as indicated by high rates of adherence and high acceptability from the AHA-NAs who completed the measures with participants. This was a positive result, as the AHA-NAs experienced staffing challenges during the pilot period and had competing workload demands, yet still achieved high participant consent rates and adherence for the participants approached. Conversely, the introduction of measures into the sarc-pathway assessment, which were not part of routine clinical practice on the acute wards, such as BIS, proved challenging for clinicians. Access to the BIS equipment was difficult on the ward, as it was located in a separate outpatient area; portability of the equipment was burdensome and transporting participants to and from the equipment was difficult, as they commonly felt fatigued, too unwell and/or declined. This was reflected by the low adherence to completing the BIS measures (only 7/34) and low acceptability from the dietitians, who were the clinicians responsible for completing these measures. Welch et al. 2021 tested the feasibility of the sarcopenia assessment in an acute older adult population and achieved a higher completion rate for bioelectrical impedance analysis (BIA) (88.2% at baseline) than our study, and recommended the use of a quadriceps ultrasound in addition to BIA as a suitable combined measure for muscle quantity and quality [34]. The use of gold standard techniques to measure muscle mass, such as computed tomography (CT), dual-energy X-ray absorptiometry (DXA) and magnetic resonance imaging (MRI), unfortunately, are not pragmatic or feasible as bedside tools [27,34]. The 5-CST was a newly introduced measure for the physiotherapists with moderate adherence (50.0%) demonstrated; however, this measure was well accepted by the clinicians.

Clinical care pathways, outcome measures and systems in the acute hospital setting for the identification, assessment and management of cancer-related malnutrition are somewhat similar but far more developed compared to those addressing the issue of cancer-related sarcopenia [6,7,18]. Cancer-related sarcopenia research has predominantly been focused on low muscle mass, and further investigations into adverse outcomes resulting from sarcopenia (including additional measures of muscle strength and function) are still warranted [2,7,35]. Highlighting the importance of sarcopenia screening, and an important finding from this study, was that 45.8% of participants ‘at risk/probable’ for sarcopenia were not identified as requiring nutrition or exercise interventions by malnutrition screening alone. Given the low completion rates of BIS in this study, a pragmatic approach may be to consider an easy and quick measure of muscle mass (i.e., calf circumference and/or BMI-adjusted calf-circumference for overweight or obese patients). Furthermore, the investigation into whether non-invasive assessment of muscle mass alongside malnutrition screening (i.e., MST) and/or existing malnutrition assessment methods by a dietitian (PG-SGA and/or Global Leadership Initiative on Malnutrition (GLIM) criteria) will further enhance referrals, and multidisciplinary interventions for patients at risk of malnutrition and sarcopenia is warranted [28,36,37]. It is logical that the integration of any new sarcopenia pathways, measures or systems should align and integrate with existing systems and the potential burden of introducing new measures should be pragmatically assessed. As described by Prado and colleagues, a patient-centred and multimodal approach is required to improve cancer patient care and outcomes in relation to malnutrition, sarcopenia and cachexia, with nutrition a critical component within a multidisciplinary framework [7]. In our study, almost half of patients on the acute cancer ward were already under the care of the dietitian and physiotherapist for other reasons and therefore gains in clinical efficiencies should be examined. Possible options could include assigning those responsible for malnutrition risk screening (and other risk screening, e.g., falls, pressure injuries) to also complete sarcopenia screening and rationalise the assessment and diagnostic tools, such as the PG-SGA, GLIM criteria and those used for muscle strength, mass and function, and to consider how referral pathways intersect between overlapping conditions, such as sarcopenia, malnutrition, cachexia, cancer-related fatigue and frailty.

Following the findings of this feasibility and acceptability study, further work should focus on adaptations to the sarc-pathway to enhance its feasibility in the acute cancer setting and acceptability to all users. Given the short hospital length of stay in the pilot ward, further consideration must to given to the ability for clinicians to assess and deliver effective interventions for participants at risk of sarcopenia whilst in hospital and ensuring appropriate post-discharge follow-up. Modifications to address the low level of re-screening for participants admitted for longer than one week are also needed. The opportunities identified by clinicians involved in the delivery of care on the sarc-pathway during the pilot, primarily relating to the use of BIS and the need for better integration with malnutrition systems, will require further attention at our cancer centre. The acute cancer ward setting provides a unique opportunity to help identify and treat cancer-related sarcopenia in an unwell population; however, consideration should be given to whether an earlier time-point in the patient cancer journey in the ambulatory setting may allow for the initiation of early and/or preventative interventions and care sooner via face-to-face clinics and/or telehealth modalities. Future opportunities for research may be to test the effectiveness of the dietitian and physiotherapist interventions implemented within the sarc-pathway in a randomised-controlled trial. Furthermore, this testing should provide greater attention to the fidelity of intervention beyond delivery, i.e., participant adherence to the dietary and exercise prescription provided.

A number of study limitations were identified. The number of participant acceptability surveys completed was small and therefore we may not have captured the views of all participants, and the discrete number of clinicians completing the survey may have led to bias, as the researchers were colleagues. The COVID-19 pandemic posed increased restrictions on the ward during the pilot period and reduced clinician access to participants and to equipment compared to pre-pandemic. In addition, exercise classes were not able to run during the pilot period, which limited the exercise prescription and treatment options we could pilot within the sarc-pathway. Limitations in the application of BIS as a valid measurement of muscle mass were acknowledged; however, this choice of bedside tool was the most accessible form of body composition equipment available at the cancer centre [27]. This study was completed with no dedicated funding and, being a ‘real-world’ pilot, reduced the number of participants that could be approached for consent for screening and relied heavily on adequate AHA-NA staffing, which was challenging within the COVID-19 environment and associated local cancer centre restrictions. Conversely, the ‘real-world’ nature of this study was a strength, as it has provided pragmatic learnings for future work in this area.

## 5. Conclusions

The sarc-pathway is a feasible and acceptable acute cancer ward clinical model that addresses cancer-related sarcopenia, albeit with opportunities for exploration, improvement of appropriate clinical assessment methods and consideration of the clinical resources required for its implementation. The sarc-pathway has provided insights and a possible framework to support the adoption of clinical guideline and position statement recommendations into clinical practice for a complex cancer population. Future work on cancer-related sarcopenia clinical models should be tailored to the setting and carefully planned. The effectiveness of such interventions should be tested in clinical practice to observe whether clinical and patient-reported outcomes can be improved for cancer patients.

## Figures and Tables

**Figure 1 ijerph-19-04038-f001:**
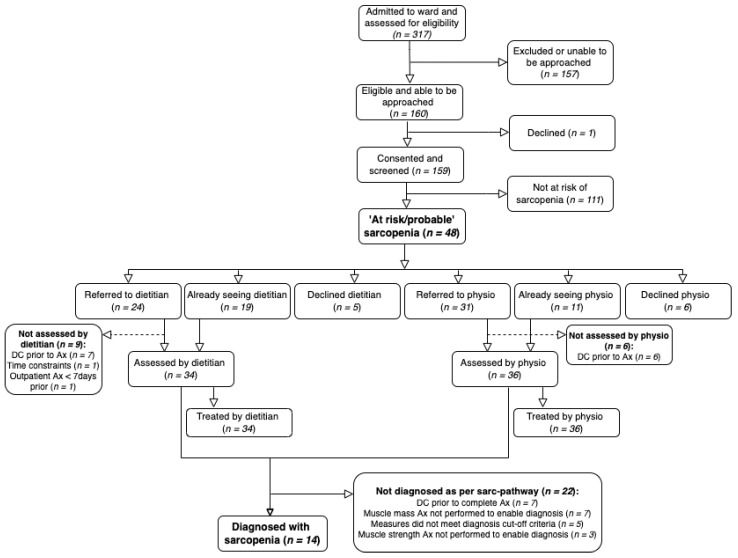
Participant flow diagram. Footnote: Abbreviations: physio = physiotherapist; DC = discharged; Ax = assessment.

**Figure 2 ijerph-19-04038-f002:**
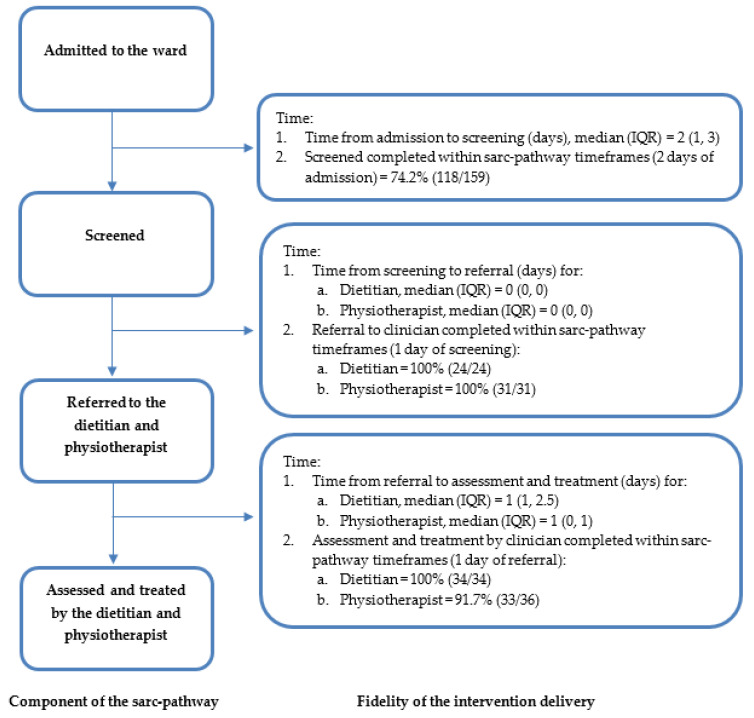
Intervention delivery adherence to key components of the sarc-pathway.

**Figure 3 ijerph-19-04038-f003:**
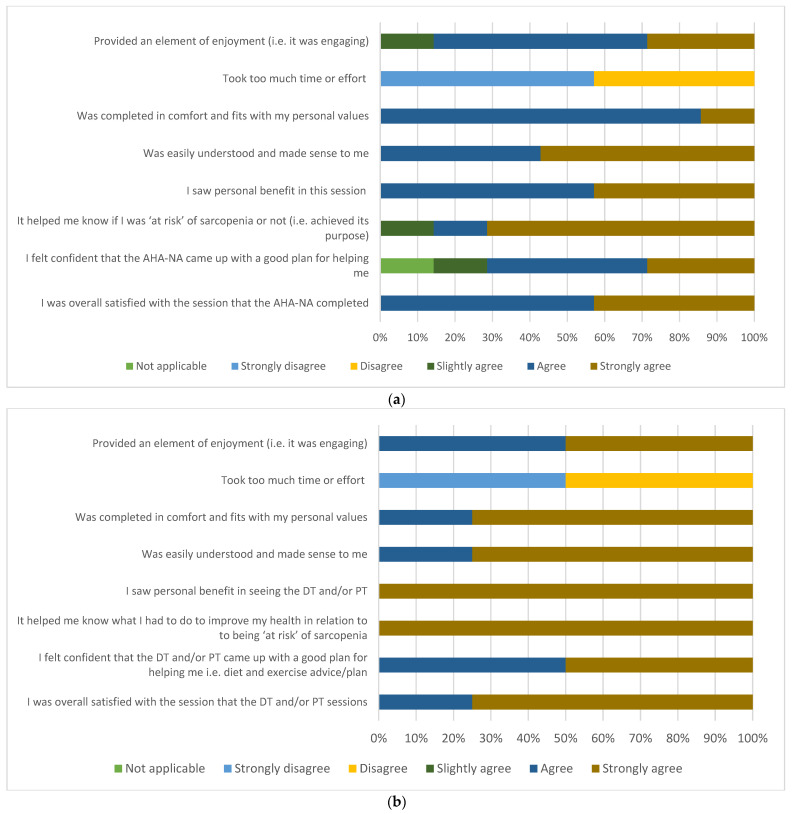
(**a**) Participant acceptability of the AHA-NA care on the sarc-pathway (*n* = 7); (**b**) participant acceptability of the dietitian and physiotherapist care on the sarc-pathway (*n* = 4). Abbreviations: AHA-NA = allied health assistant–nutrition assistant; DT = dietitian; PT = physiotherapist.

**Figure 4 ijerph-19-04038-f004:**
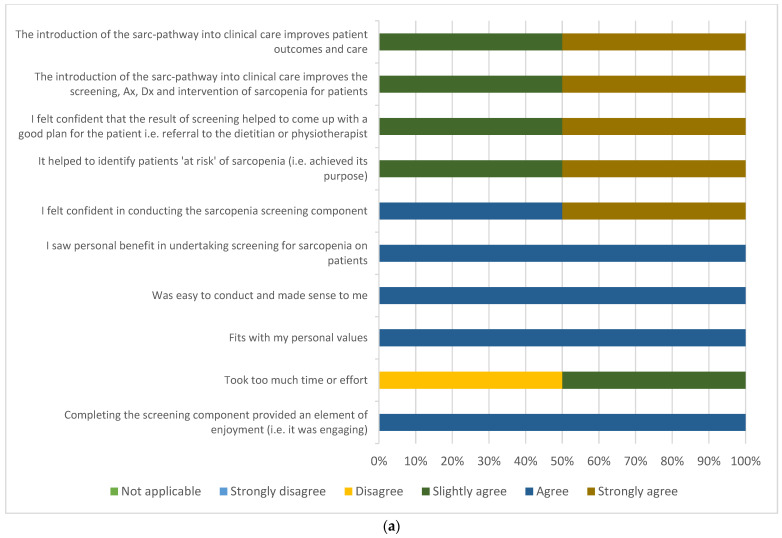
(**a**) Clinician (AHA-NA) acceptability of the sarc-pathway (*n* = 2); (**b**) clinician (dietitian, *n* = 4 and physiotherapists/AHA-PT, *n* = 6) acceptability of the sarc-pathway. Abbreviations: Ax = assessment; Dx = diagnosis; DT = dietitian; PT = physiotherapist; AHA-PT = allied health assistant–physiotherapist.

**Table 1 ijerph-19-04038-t001:** Component behaviours of the sarcopenia pathway, described in accordance with the AACTT framework ^1^.

Action:	Conduct sarcopenia screening (and re-screening) i.e., SARC-CalF, HGS	Provide written information to all participants screened (both at risk and at low risk of sarcopenia)	Refer participants at risk of sarcopenia to the dietitian and physiotherapist	Complete full individualised assessment with participants	Complete clinical assessment measures (for dietitian and physiotherapist assessments and diagnosis of sarcopenia, i.e., PG-SGA, BIS, 5-CST, AKPS)	Deliver individualised interventions	Where indicated, deliver outpatient care and/or refer to external services
Actor:	AHA-NA	AHA-NA	AHA-NA	Dietitian and physiotherapist	AHA-NA, dietitian, physiotherapist (may be delegated to AHA-PT)	Dietitian, physiotherapist (may be delegated to AHA-PT)	Dietitian and physiotherapist
Context:	Acute cancer inpatient ward	Acute cancer inpatient ward	Acute cancer inpatient ward—referral via EMR	Acute cancer inpatient ward—participants’ room and/or ward	Acute cancer inpatient ward—participants’ room, ward and/or gym	Acute cancer inpatient ward—participants’ room, ward and/or gym	Clinic room, gym, via telehealth and/or via external provider
Target:	Eligible participants admitted to the ward (and those screened as low risk of sarcopenia on admission and still an inpatient at day 7)	All participants screened (both at risk and at low risk of sarcopenia)	Participants considered at risk of sarcopenia after screening	Participants considered at risk of sarcopenia after screening	Participants considered at risk of sarcopenia after screening and undertaking assessment by the dietitian and physiotherapist	Participants considered at risk of sarcopenia after screening and/or diagnosed with sarcopenia	Participants considered at risk of sarcopenia after screening and/or diagnosed with sarcopenia requiring ongoing intervention post discharge
Time *:	Within 2 days of admission for initial screen (day 6–8 for rescreen)	Within 2 days of admission	Within 2 days of admission	Within 1 day of referral being placed via the EMR	Baseline measures—AHA-NA: within 2 days of admission; dietitian/ physiotherapist within 1 day of referral; pre-discharge measures—1–2 days prior to hospital discharge by dietitian/physiotherapist	Within 1 day of referral from NA and then as specified by dietitian/physiotherapist	Following discharge from hospital

Note: ^1^ Presseau, J.; McCleary, N.; Lorencatto, F.; Patey, A.M.; Grimshaw, J.M.; Francis, J.J. Action, actor, context, target, time (AACTT): A framework for specifying behaviour. *Implement Sci*. **2019**, *14*, 102. * times do not include weekend days. Abbreviations: SARC-CalF = sarcopenia screening tool including Sarc-F tool and calf circumference measurement; HGS = handgrip strength; AHA-NA = allied health assistant–nutrition assistant; AHA-PT = allied health assistant–physiotherapy; PG-SGA = Patient-Generated Subjective Global Assessment; BIS = bioimpedance spectroscopy; 5-CST = 5-times chair stand test; AKPS = Australia-modified Karnofsky Performance Status scale; EMR = electronic medical record.

**Table 2 ijerph-19-04038-t002:** Demographic and clinical characteristics of participants who consented to care on the sarc-pathway.

Characteristics	Participants on the Sarc-Pathway (*n* = 159)
Age (years), median (IQR)	61 (49, 70)
Gender (male)	89 (56.0)
Cancer diagnosis:	
Sarcoma	27 (17.0)
Lung	26 (16.4)
Lower gastrointestinal	19 (11.9)
Skin/melanoma	18 (11.3)
Upper gastrointestinal	14 (8.8)
Genitourinary	13 (8.2)
Head and neck	11 (6.9)
Haematological	10 (6.3)
Cervical/ovarian	10 (6.3)
Breast	9 (5.7)
Brain and spine	2 (1.3)
Length of hospital stay (days), median (IQR)	5 (3, 7)

Note: values are reported as *n* (%) unless stated.

**Table 3 ijerph-19-04038-t003:** Adherence to key ‘time’ components of the sarc-pathway.

Component of the Sarc-Pathway	*n* = 159
Screening within sarc-pathway timeframes (2 days of admission):	
Completed	118 (74.2)
Not completed:	41 (25.8)
Missed by AHA-NA	19 (11.9)
Weekend (no AHA-NA screening)	15 (9.4)
Transferred from other ward—delay	2 (1.3)
No AHA-NA staffing available	2 (1.3)
COVID-19 restrictions, i.e., isolation requirements for participant	2 (1.3)
Participant medically unstable	1 (0.6)
Referral to dietitian (*n* = 24)	
Referral to dietitian completed within sarc-pathway timeframes (1 day of screening):	
Completed	24 (100)
Not completed	0 (0)
Referral to physiotherapist (*n* = 31)	
Referral to physiotherapist completed within sarc-pathway timeframes (1 day of screening):	
Completed	31 (100)
Not completed	0 (0)
Assessment and treatment by dietitian (*n* = 34), i.e., those referred + already being seen	
Assessment and treatment by dietitian completed within sarc-pathway timeframes (1 day of referral):	
Completed	34 (100)
Not completed	0 (0)
Assessment and treatment by physiotherapist (*n* = 36), i.e., those referred + already being seen	
Assessment and treatment by physiotherapist completed within sarc-pathway timeframes (1 day of referral):	
Completed	33 (91.7)
Not completed:	3 (8.3)
COVID-19 precautions, i.e., isolation requirements for participant	1 (2.8)
Time delays due to competing priorities	1 (2.8)
Known already to clinician and clinical measures not collected as per sarc-pathway	1 (2.8)

Note: values are reported as *n* (%). Abbreviations: AHA-NA = allied health assistant–nutrition assistant.

**Table 4 ijerph-19-04038-t004:** Adherence to and scores of the clinical screening and assessment measures within the sarc-pathway.

Clinical Assessment Measures	Participants	Score/Outcome
*Screening (n = 159)*		
Hand Grip Strength (Maximum), kg		
Completed	159 (100.0)	28 (20, 37)
Not completed:	0 (0.0)	
SARC-F Score		
Completed	159 (100.0)	2 (1, 3)
Not completed:	0 (0.0)	
Calf Circumference (Maximum), cm		
Completed	159 (100.0)	36.8 (5.8)
Not completed:	0 (0.0)	
SARC-CalF Score		
Completed	159 (100.00)	3.0 (1, 10)
Not completed:	0 (0.0)	
*Assessment by dietitian (n = 34)*		
ALM via BIS, kg		
Completed	7 (20.6)	14.6 (2.2)
Not completed:	27 (79.4)	
Patient declined but otherwise able	8 (23.5)	
Patient unable due to medical/ physical limitations	6 (17.6)	
Discharged before completed	6 (17.6)	
Not attempted	2 (5.9)	
Change in patient medical condition	2 (5.9)	
Equipment issue	1 (2.9)	
COVID-19 precautions	1 (2.9)	
Patient became fatigued	1 (2.9)	
PG-SGA Score		
Completed	26 (76.5)	12.7 (4.9) PG-SGA A, *n* (%) = 6 (23.1) PG-SGA B, *n* (%) = 16 (61.5) PG-SGA C, *n* (%) = 4 (15.4)
Not completed:	8 (23.5)	
Missed by clinician	4 (11.8)	
Discharged before completion	3 (8.8)	
COVID-19 precautions	1 (2.9)	
*Assessment by physiotherapist (n = 36)*		
5-CST, seconds		
Completed	18 (50.0)	17.5 (12.7, 23.3)
Not completed:	18 (50.0)	
Patient unable due to medical/ physical limitations	6 (16.7)	
Not attempted	5 (13.9)	
Discharged before completion	3 (8.3)	
COVID-19 precautions	1 (2.8)	
Stopped mid-test	1 (2.8)	
Change in patient medical condition	1 (2.8)	
Missed	1 (2.8)	
AKPS Score		
Completed	26 (72.2)	100, *n* (%) = 0 (0) 90, *n* (%) = 1 (3.8) 80, *n* (%) = 3 (11.5) 70, *n* (%) =3 (11.5) 60, *n* (%) = 11 (42.3) 50, *n* (%) = 3 (11.5) 40, *n* (%) = 4 (15.4) 30, *n* (%) = 1 (3.8) ≤20, *n* (%) = 0 (0)
Not completed:	10 (27.8)	
Missed	10 (27.8)	

Note: values are reported as *n* (%), median (IQR) or mean (SD) as appropriate. Abbreviations: SARC-F = strength, assistance in walking, rise from chair, climb stairs and falls; SARC-CalF = SARC-F tool and calf circumference measurement; ALM = appendicular lean mass; BIS = bioimpedance spectroscopy; PG-SGA = Patient-Generated Subjective Global Assessment; 5-CST = Five-times chair stand test; AKPS = Australia-modified Karnofsky Performance Status scale.

**Table 5 ijerph-19-04038-t005:** Summary of participants’ sarcopenia and malnutrition risk and diagnostic status.

Clinical Characteristics	Participants on the Sarc-Pathway
Sarcopenia risk (*n* = 159):	
At risk or probable sarcopenia *	48 (30.2)
Not at risk	111 (69.8)
Sarcopenia diagnosis * (*n* = 26):	
Yes	14 (8.8)
No	22 (13.8)
Malnutrition risk (*n* = 159):	
At risk (MST ≥ 2)	62 (39.0)
Not at risk (MST < 2)	97 (61.0)
Malnutrition diagnosis (*n* = 26):	
Yes (PG-SGA category B or C)	20 (12.6)
No (PG-SGA category A)	6 (3.8)

Note: values are reported as *n* (%) and median (IQR). * refer to Appendix A for definitions and clinical cut-offs. Abbreviations: MST = Malnutrition Screening Tool; PG-SGA = Patient-Generated Subjective Global Assessment.

## Data Availability

Data conducted as part of the study can be requested via the corresponding author for any reasonable request.

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
