# Peer review of "Development and Feasibility of an Inpatient Cancer-Related Sarcopenia Pathway at a Major Cancer Centre"

_ijerph, 2022, doi:10.3390/ijerph19074038_

Round 1

Reviewer 1 Report

The manuscript has a good design, execution and writing.

A consideration for future interventions would be to include a professional from physical activity and sports sciences, and not just a physiotherapist and exercise physiologist for the physical part.

Some errors that should be corrected in the text are the following:

Line 146: There is no mention to Figure 1 in the paragraph.

Line 320: There is a mistake with the title referencing Fig. 2. It is not correct the title “Supplementary Material File 3 . Participant flow diagram”.

Line 350:  There is an error naming Figure 3 as 2, entitled “Figure 2. Intervention delivery adherence to key components of the sarc-pathway”.

Congrats!

Reviewer 2 Report

The manuscript is well written and represents a needed knowledge in the field. Only few comments to optimize the manuscript's quality as follows:

  • Please provide full pages numbering to the following references: 8, 9, 13, 19, 20, 29, 34.
  • Figure 2 is recommended to be represented in a clear format with black/white background.
  • Please use a landscape format for Supplementary Material File 3 . Participant flow diagram.

Reviewer 3 Report

First, congratulations for your study. I comment some questions to improve this manuscript.

INTRODUCTION:

Introduction is very correct, but relationship with sarcopenia you should add some comment about EWGSOP 2, for improve your introduction.

METHODS:

What test did you use to control the cognitive impairment in patients? You should add the validate tool to control cognitive impairment.

Results and Discussion are perfect to understand this manuscript.

Good work, congratulations.
